# Implementing postpartum family planning services in rural Rwanda: A mixed-methods study

Claudine Umuhoza[1], Mame Diarra Niang[2], Rosine Ingabire[1], Julien Nyombayire[1], Amelia Mazzei[1,3], Rachel Parker[3], Amanda Tichacek[3], Azhar Nizam[4], Jessica M. Sales[5], Lisa B. Haddad[6], Phaedra Corso[7], Susan Allen[3], Etienne Karita[1,3], Kristin M. Wall[2]*

1 Projet San Francisco (PSF)/Center for Family Health Research (CFHR), Kigali, Rwanda, 2 Department of Epidemiology, Rollins School of Public Health, Emory University, Atlanta, Georgia, United States of America, 3 Department of Pathology, School of Medicine, Emory University, Atlanta, Georgia, United States of America, 4 Department of Biostatistics and Bioinformatics, Rollins School of Public Health, Emory University, Atlanta, Georgia, United States of America, 5 Department of Behavioral, Social and Health Education Sciences, Rollins School of Public Health, Emory University, Atlanta, Georgia, United States of America, 6 Population Council, Center for Biomedical Research, New York, New York, United States of America, 7 Office of the Vice Chancellor for Research, Indiana University, Indianapolis, Indiana, United States

* kmwall@emory.edu

## Abstract

### Introduction

Postpartum family planning (PPFP) reduces adverse maternal-child outcomes related to short interpregnancy intervals and unintended pregnancies. This mixed-method study assessed PPFP needs in rural government health facilities as well as clients' knowledge and barriers to PPFP uptake in Rwanda.

### Methods

From May-July 2023, we conducted cross-sectional PPFP needs assessments in rural government health facilities and focus group discussions (FGDs) among couples attending antenatal clinics to understand women's and men's perceptions and barriers to PPFP uptake. Quantitative data were collected from twelve rural government health facilities (two hospitals, four health centers, six health posts). Qualitative data were collected during six FGD with 6–12 participants per session. Quantitative data were analyzed descriptively, and qualitative data were analyzed thematically with a deductive approach.

### Results

Seventeen (65%) hospital nurses and 11 (23%) health center nurses were trained in implant insertion, and six (23%) hospital nurses and four (9%) health center nurses were trained in postpartum intrauterine device (PPIUD) insertion. Hospitals provided an average of 204 postpartum implants (29% of deliveries) and seven PPIUDs per

**Data availability statement:** Wall, Kristin, 2025, "Replication Data for: Implementing postpartum family planning services in rural Rwanda: a mixed-methods study", https://doi.org/10.7910/DVN/0EVX40, Harvard Dataverse.

**Funding:** National Institutes of Health (https://nih.gov/) R01HD101600 to KMW. Support was also available from the Emory University Global Field Experience program (https://sph.emory.edu/rollins-life/community-engaged-learning/global-field-experience/index.html) to MDN. The funders had no role in the study design; collection, management, analysis, or interpretation of data; writing of the report; or the decision to submit the report for publication.

**Competing interests:** The authors have declared that no competing interests exist.

month (1% of deliveries), while health centers provided 25 postpartum implants and no PPIUDs per month. At health posts, there was no equipment for implant or intrauterine device (IUD) provision. FGD findings revealed that couples have access to family planning counseling at the health center, but they were concerned about limited information on contraceptive method mechanisms of action and side effects; knowledge about and access to IUD/PPIUD was especially limited.

## Conclusions

Enhanced PPFP training and provision is needed in rural areas, especially for PPIUD. Knowledge gaps and concerns about side effects were emphasized in FGDs. PPFP demand creation strategies tailored for the rural populace as well as rural provider training could improve PPFP access and uptake in rural government clinics of Rwanda.

## Background

The World Health Organization (WHO) recommends postpartum family planning (PPFP) for birth spacing, prevention of unintended pregnancy and abortion, and maternal-child health [1–3]. PPFP positively impacts women, children, and nations. Short birth spacing intervals are associated with preterm birth, low birth weight, small gestational age, and maternal-child death [2,4]. It is estimated that PPFP reduces annual maternal deaths by 30% and child deaths by 10% in developing countries by preventing unintended pregnancy and spacing pregnancies ≥2 years as recommended by the WHO [2,5–7].

Long-acting reversible contraceptives (LARC) are highly effective, cost-effective, feasible, acceptable [8–11], and safe for postpartum use [1,2]. LARCs include the copper intrauterine device (IUD) and implant which last >12 and >5 years and cost $0.40 and $9 USD [12], respectively. In contrast, shorter-acting pills and injectables have lower effectiveness [2] and are refilled monthly or 3-monthly at a cost of $1 USD/cycle [12]. In Rwanda, all women and men, including adolescents and young adults, have access to family planning services through both public and private health institutions. These services are covered by public health insurance for a majority of clients under a universal healthcare system. Short-term family planning methods may provided free of charge by community health workers (CHW) following initiation at a health facility [13].

PPFP and LARC are prioritized in the Rwanda Family Planning 2030 Commitment [14,15]. However, similar to other countries in the region (Uganda, Burundi, Kenya, and Tanzania), 51% of married Rwandan women have unmet PPFP need, and this unmet need is more than two times higher compare to non-postpartum periods [16,17]. LARC use also remains low: 27% of Rwandan women use implants and 1.2% use IUDs [17].

PPFP could have a major life-saving impact for rural Rwandans. Most (80%) Rwandans live in rural areas [17]. Though desires for large family sizes in rural areas have decreased due to government educational campaigns on health, poverty, and land

scarcity [18], rural total fertility remains higher compared to urban areas (4.3 vs 3.4) despite the mean desired number of children being similar to urban areas (3.5 vs 3.4). Though rural areas have seen a 39% increase in family planning use over the past decade [17,19], rural women report lower IUD use (0.7% vs 3%) than urban women [17]. Additionally, inter-pregnancy intervals are shorter and rates of infant, under-5 [17], and maternal [20] mortality are higher in rural vs urban women.

Rural women/couples face distinct challenges and opportunities to access PPFP. Challenges include more poverty and healthcare access barriers, lower education and literacy, and longer distances to health facilities [17]. A recent study found travel distance made it difficult for rural Rwandan women to attend all four recommended antenatal care (ANC) visits [21], and rural women have concerns about transportation costs [22]. Studies conducted in Rwanda [23–26] and other countries in the region [27,28] show limited male partner involvement in family planning counseling. Similar to women living in urban areas, most (88%) rural women report family planning is a decision between husband and wife [17], though engaging men can be challenging. In our prior study in 30 rural Rwandan health centers, 77% of facility directors and family planning providers expressed concerns about engaging men in family planning decisions due to men not being willing to wait at the facility and miss income opportunities [29].

There are several opportunities to access PPFP within Rwanda's unique healthcare system. A robust CHW network provides FP and PPFP educational counseling in the community, and also refers and sometimes accompanies clients to health centers or health posts to receive care. CHW can also provide short-term contraceptive methods after initiation at a health facility. Health posts are Rwanda's most basic healthcare facilities and are overseen by health centers in the same patient catchment area. Health posts provide a limited range of services such as outpatient care, child immunizations, growth monitoring for children under the age of five, antenatal care, family planning counseling and provision, health education, and limited laboratory tests (malaria rapid testing, HIV testing, pregnancy testing, and glycemia/glycosuria rapid testing). Health posts refer patients to health centers, and when necessary, to district hospitals. Health centers provide family planning counseling and provision, preventive and therapeutic healthcare, and referrals to district hospitals if necessary. Most health centers are staffed by nurses and midwives but not doctors (except for medicalized health centers). Hospitals do not provide antenatal care (ANC) or infant vaccination services but do provide FP/PPFP services. [29].

Although limited, recent studies of women residing in rural areas have confirmed low PPFP uptake [30,31], but studies have not evaluated the facility readiness to offer PPFP services nor explored rural women's and male partners PPFP needs and decision-making.

Our previous study assessing FP program needs in 30 rural health centers, has highlighted lack of adequate FP equipment, trained staff especially in LARC provision as well as lack of CHW awareness and demand creation strategies, which could be a significant barrier to FP access. [29]. In order to better understand PPFP knowledge and uptake barriers in rural Rwanda, we conducted focus group discussions (FGDs) with couples and evaluated the resources and gaps for PPFP provision in rural health facilities to improve PPFP program implementation in rural areas.

## Methods

### Study design

This study used a concurrent mixed methods design, including quantitative data from a cross-sectional needs assessment survey and qualitative data from couple FGDs. Following the convergent mixed methods design described by Cresswell and Clark, 2011 [32], in which the quantitative and qualitative data are collected and analyzed concurrently, and the data are then interpreted together to better contextualize and understand challenges and opportunities for PPFP implementation in rural Rwanda.

### Study sites and recruitment

The study was conducted from May-July 2023 in two rural districts in Rwanda, Rusizi and Rubavu. These districts are located in the Western Province, which has the highest fertility rate and unmet need for FP in Rwanda [17,33]. Our study

team previously worked in 14 government health facilities in these districts to implement an Ebola vaccination campaign and vaccination safety trial in pregnant women [34–36]. In the present study, we selected from these 14 government health facilities with which we had prior relationships for concurrent PPFP needs assessment and FGD recruitment. We selected the 12 facilities which were of highest volume, geographically accessible by car, and non-Catholic (as Catholic facilities are not required to provide family planning counseling or methods, though their adjacent health posts do provide family planning educational counseling).

Twelve facilities were selected: one district hospital, two affiliated health centers, and three affiliated health posts per district.

FGDs recruited 27 couples using convenience sampling from ANC clinics within the four selected health centers (only health centers conduct routine ANC visits; hospitals/health posts do not provide ANC services). Eligible couples were comprised of pregnant women attending ANC visits with their partners, both partners were at least 18 years of age, both were willing to participate as couples in the FGDs, and both provided signed informed consent prior to FGDs.

## Quantitative data collection and analysis: PPFP needs assessments

To better understand the current status of and infrastructure for PPFP service delivery in rural Rwanda, we conducted facility needs assessments in the 12 health facilities using a standardized questionnaire adapted from prior needs assessments used for assessing family planning program infrastructure in Rwanda [29]. The needs assessments evaluated facility volume in different maternal-child services, the human resource capacity to provide PPFP services, and whether facilities had materials available for PPFP educational counseling and service provision. For the facility personnel, we recorded the number of available and trained providers (nurses, midwives, and physicians) in PPFP promotion and provision, and the number of affiliated and trained CHW in PPFP/Postpartum Long Acting Reversible Contraception (PPLARC) educational counseling in the community [29]. We also extracted facility-level data over six months (November 2022-April 2023) from monthly facility reports: the facility catchment population and monthly numbers of women attending ANC, labor and delivery, and PPFP services, and the type of PPFP provided.

The needs assessment questionnaire was administered by the Center for Family Health Research (CFHR) interview team, consisting of two trained nurses and one intern student. The team first met with each facility's key stakeholders, including the facility director, heads of departments, and the data manager. Interviews were conducted in Kinyarwanda.

No personal identifiers were collected. Needs assessment data were entered into a Microsoft Access database, cleaned, and analyzed descriptively (using counts and percentages) using Excel.

## Qualitative data collection and analysis: FGDs

We used a standardized focus group guide developed in English based on prior studies, predefined priority areas, and observation of PPFP services on the ground in order to better describe PPFP services in rural areas. The FGD guide assessed participants' access to FP counseling and current FP knowledge, PPFP knowledge, perceived PPFP barriers and proposed solutions, and potential suggested amendments to a PPFP educational flipchart already in use in several health facilities in the Rwandan capital [23,37–40]. The final guide was translated into Kinyarwanda and pilot-tested.

The FGDs were conducted in July 2023 at the selected rural health centers by two female CFHR nurses (Registered Nurses), who were trained about the study and experienced in conducting FGDs, with one nurse moderating the FGD and the other taking notes during the discussions. The participants were recruited from antenatal care services at the selected health centers and were provided with information about the study purpose and procedures prior to providing informed consent. After consenting, all couples were gathered in one room. Wives and husbands were encouraged to talk at any time. Questions specifically targeting wives or husbands only were also asked in the same room without separating partners so as to maintain a flow of conversation. These discussions were audio-recorded to avoid missing any participant views during the transcription process. We conducted six FGDs with 6–12 participants per session with a time duration

ranging between 1 hour 45 minutes to 2 hours 15 minutes, with three FGDs in each hospital catchment area. After completion of an FGD, notes/recordings were translated from Kinyarwanda to English by CFHR nurses, and the transcripts were imported into NVivo software to group the responses to each question using the participants' own words.

We conducted a qualitative content analysis with a deductive approach which involves starting with pre-defined themes derived from prior research or theory [41]. Our focus group guides were adapted from and used pre-specified themes from our prior published formative work with women and couples to understand client perspectives of PPFP in Kigali, the capital of Rwanda [23,39]; those guides were developed grounded in the Theory of Planned Behavior [42]. Data were grouped by these pre-specified themes and sub-themes (see S1 Fig) and representative quotes for each, inclusive of majority and minority participants and positive and negative views, were selected for illustrative purposes. Such deductive thematic analysis applies pre-defined themes derived from existing theories or prior research to qualitative data, without explicitly assigning codes to segments of text [43]. We also allowed space for inductive themes to emerge from the discussions, and data collection continued until thematic saturation was reached. Transcripts were not returned to the participants, and participant feedback was not solicited on the findings.

### Ethics

We used de-identified programmatic data shared by the government facility's key stakeholders during the needs assessments. Ethical approval to conduct FGDs was obtained from the Rwanda National Ethics Committee and Emory Institutional Review Board, and written informed consent from eligible couples was obtained before allowing them to participate in any FGD, and each participant was compensated for the time.

### Results

#### Human resources and equipment for PPFP

The two district hospitals served a catchment population size of 737,233 across 731 villages while the four affiliated health centers served a subset catchment population size of 186,508 across 172 villages (Table 1). A total of 2,808 CHW were affiliated with hospitals of whom 571 (20%) were in charge of pregnant women and newborn health. Another 633 CHW were affiliated with the health centers of whom 26% were in charge of pregnant women and newborn health, 14% were in charge of family planning, 59% were trained in PPFP promotion, and 51% were trained in PPLARC promotion.

Of the 206 nurses working at hospitals, 26 were assigned to maternity departments, of whom 65% were trained in PPFP promotions, 27% were trained in IUD insertion, 23% were trained in PPIUD insertion, and 65% were trained in implant insertion. Among 44 midwives working at hospitals and assigned in maternity departments, 82% were trained in PPFP promotion, 16% were trained in IUD and PPIUD insertion, and 77% were trained in implant insertions. Of the 47 nurses working at health centers, 36% were trained in PPFP promotions, 15% were trained in IUD insertion, 9% were trained in PPIUD insertion, and 23% were trained in implant insertion. Of five midwives at health centers, 40% were trained in PPFP promotion, IUD, PPIUD, and implant insertion. Only hospitals employed physicians; of 22 physicians working in maternity department, 32% were trained in PPFP promotion, IUD insertion, PPIUD insertion (including intracesarean PPIUD insertion), and 14% were trained in implant insertion.

All six facilities reported available LARC methods in stock, implant disposable insertion kits, equipment for LARC insertion (forceps, specula, and gynecological exam tables), the capacity to sterilize IUD insertion kits either by an autoclave or dry heat sterilization, and family planning educational counseling materials. However, few gynecologic lamps were available and only half of the health centers had a working television and generator.

#### PPLARC provision

The two hospitals performed an average of 704 deliveries per month and provided 204 PP implants, seven PPIUD, and three PPIUD follow-up appointments (Table 2). Almost one-third (30%) of women delivering at hospitals received

**Table 1. Human resources and equipment for PPFP in rural-serving health facilities.**

| | Hospitals | | | | Health Centers | | | | | |
|---|---|---|---|---|---|---|---|---|---|---|
| | Total | | Gisenyi | Gihundwe | Total | | Kigufi | Rugerero | Bugarama | Giheke |
| | N | % | N | N | N | % | N | N | N | N |
| Catchment population | 737,233 | – | 546,683 | 190,550 | 186,508 | – | 48,718 | 66,926 | 49,207 | 21,657 |
| Villages served | 731 | – | 518 | 213 | 172 | – | 48 | 51 | 37 | 36 |
| Total CHW | 2,808 | – | 1,990 | 818 | 663 | – | 192 | 179 | 148 | 144 |
| In charge of pregnant women and newborn health | 571 | 20% | 358 | 213 | 172 | 26% | 48 | 51 | 37 | 36 |
| In charge of family planning | – | – | – | – | 94 | 14% | 46 | 33 | 9 | 6 |
| Trained in PPFP promotion | – | – | – | – | 388 | 59% | 96 | 0 | 148 | 144 |
| Trained in PPFP IUD/implant promotion | – | – | – | – | 338 | 51% | 46 | 0 | 148 | 144 |
| Total nurses | 206 | – | 121 | 85 | 47 | – | 10 | 16 | 12 | 9 |
| Total nurses in maternity* department at hospitals | 26 | – | 17 | 9 | 47 | – | 10 | 16 | 12 | 9 |
| Trained in PPFP promotions | 17 | 65% | 10 | 7 | 17 | 36% | 1 | 3 | 2 | 5 |
| Trained in IUD insertion | 7 | 27% | 4 | 3 | 7 | 15% | 1 | 2 | 2 | 2 |
| Trained in PPIUD insertion | 6 | 23% | 4 | 2 | 4 | 9% | 1 | 0 | 2 | 1 |
| Trained in implant insertion | 17 | 65% | 10 | 7 | 11 | 23% | 1 | 3 | 2 | 5 |
| Total midwives | 47 | – | 29 | 18 | 5 | – | 1 | 1 | 1 | 2 |
| Assigned in maternity | 44 | – | 29 | 15 | 5 | – | 1 | 1 | 1 | 2 |
| Trained in PPFP promotions | 36 | 82% | 29 | 7 | 2 | 40% | 1 | 0 | 0 | 1 |
| Trained in IUD insertion | 7 | 16% | 2 | 5 | 2 | 40% | 1 | 0 | 0 | 1 |
| Trained in PPIUD insertion | 7 | 16% | 2 | 5 | 2 | 40% | 1 | 0 | 0 | 1 |
| Trained in implant insertion | 34 | 77% | 29 | 5 | 2 | 40% | 1 | 0 | 0 | 1 |
| Total physicians in maternity department | 22 | – | 10 | 12 | – | – | – | – | – | – |
| Trained in PPFP promotion | 7 | 32% | 4 | 3 | – | – | – | – | – | – |
| Trained in IUD insertion | 7 | 32% | 4 | 3 | – | – | – | – | – | – |
| Trained in PPIUD insertion | 7 | 32% | 4 | 3 | – | – | – | – | – | – |
| Trained in implant insertion | 3 | 14% | 0 | 3 | – | – | – | – | – | – |
| Trained in intra-cesarean PPIUD insertion | 7 | 32% | 4 | 3 | – | – | – | – | – | – |
| Equipment amounts available (number) | | | | | | | | | | |
| IUD methods | 235 | – | 220 | 15 | 106 | – | 23 | 32 | 36 | 15 |
| Implants methods | 379 | – | 322 | 57 | 187 | – | 70 | 95 | 14 | 8 |
| Disposable Jadelle kits | 333 | – | 322 | 11 | 184 | – | 70 | 95 | 11 | 8 |
| Non-disposable Jadelle kits | 0 | – | 0 | 0 | 0 | – | 0 | 0 | 0 | 0 |
| Forceps | 50 | – | 6 | 44 | 47 | – | 3 | 14 | 9 | 21 |
| Speculum | 21 | – | 6 | 15 | 37 | – | 6 | 4 | 10 | 17 |
| Sims vaginal speculum | 17 | – | 4 | 13 | 13 | – | 1 | 6 | 6 | 0 |
| Hysterometer | 11 | – | 7 | 4 | 8 | – | 2 | 5 | 1 | 0 |
| Tenaculum forceps | 11 | – | 4 | 7 | 9 | – | 3 | 2 | 3 | 1 |
| Kelly forceps | 12 | – | 3 | 9 | 17 | – | 1 | 8 | 4 | 4 |
| Gynecological tables | 13 | – | 8 | 5 | 10 | – | 2 | 3 | 2 | 3 |
| Gynecological lamps | 3 | – | 2 | 1 | 2 | – | 1 | 0 | 0 | 1 |
| String scissors | 17 | – | 14 | 3 | 8 | – | 1 | 3 | 2 | 2 |
| IUD/Implants kits able to be sterilized per day | 30 | – | 18 | 12 | 26 | – | 10 | 12 | 2 | 2 |
| Equipment presence | | | | | | | | | | |
| Working generator | – | – | Yes | Yes | – | – | Yes | No | No | No |
| Autoclave | – | – | Yes | Yes | – | – | Yes | Yes | No | Yes |
| Dry heat sterilization | – | – | No | No | – | – | Yes | Yes | Yes | No |

*(Continued)*

**Table 1.** (Continued)

| | Hospitals | | | | Health Centers | | | | | |
|---|---|---|---|---|---|---|---|---|---|---|
| | Total | | Gisenyi | Gihundwe | Total | | Kigufi | Rugerero | Bugarama | Giheke |
| | N | % | N | N | N | % | N | N | N | N |
| Television | – | – | Yes | Yes | – | – | Yes | No | No | No |
| Family planning educational counseling materials | – | – | Yes | Yes | – | – | Yes | Yes | Yes | Yes |
| NGO or other partner in PPFP or PPIUD | – | – | No | Yes | – | – | No | No | Yes | No |

CHW: Community health workers, NGO: non-governmental organization, PPIUD: postpartum intrauterine device, IUD: intrauterine device, PPFP: post-partum family planning, *Or ANC for health centers.

**Table 2.** Monthly average service volume and PPLARC provision from November 2022 to April 2023 in rural-serving health facilities.

| | Hospitals | | | Health Centers | | | | |
|---|---|---|---|---|---|---|---|---|
| | Total | Gihundwe | Gisenyi | Total | Kigufi | Rugerero | Bugarama | Giheke |
| | N | N | N | N | N | N | N | N |
| Facility service attendees | | | | | | | | |
| Infant vaccination attendees* | – | – | – | 2058 | 638 | 472 | 690 | 258 |
| Antenatal care attendees* | – | – | – | 320 | 88 | 83 | 100 | 49 |
| Deliveries | 704 | 224 | 480 | 145 | 24 | 35 | 65 | 21 |
| Method provision | | | | | | | | |
| PPLARC total | 210 | 39 | 171 | 25 | 7 | 11 | 6 | 2 |
| Postpartum implant insertions | 204 | 36 | 168 | 25 | 7 | 11 | 6 | 2 |
| PPIUD insertions | 7 | 4 | 3 | 0 | 0 | 0 | 0 | 0 |
| PPIUD Follow Up | 3 | 3 | 0 | 0 | 0 | 0 | 0 | 0 |
| Interval (non-postpartum) IUD insertions | 11 | 6 | 5 | 5 | 1 | 2 | 0 | 3 |
| | % | % | % | % | % | % | % | % |
| PPLARC insertions per delivery | 30% | 18% | 36% | 17% | 28% | 30% | 8% | 8% |
| Postpartum implant insertions per delivery | 29% | 16% | 35% | 17% | 28% | 30% | 8% | 8% |
| PPIUD insertions per delivery | 1% | 2% | 1% | 0% | 0% | 0% | 0% | 0% |

*Services provided at health centers only, PPIUD: postpartum intrauterine device, PPLARC: postpartum long-acting reversible contraception, IUD: intrauterine device.

a PPLARC (29% implant, 1% IUD). The four health centers performed 145 deliveries per month and provided 25 PP implants and no PPIUDs. Almost one-fifth (17%) of women delivering at health centers received a PPLARC (17% implant, 0% IUD).

Among the six health posts, only one was providing delivery services with an average of 18 deliveries monthly (data not tabled). There was no available equipment to facilitate implant and IUD provision, and no PPLARC provision was performed. All health posts were only providing short-term contraceptive methods.

## Couple focus group discussions: qualitative data

We conducted six FGDs which included a total of 27 couples (Table 3). The average age for women was 28 years, ranging from 20 to 43 years, while the average age for men was 31 years, ranging from 21 to 58. The average number of pregnancies including the current pregnancy was three, with a range of one to seven. Sixty-three percent of women and 85% of men had completed primary education only, and an additional 15% of women did not complete any education.

**Table 3. Characteristics of focus group participants (N = 27 pregnant couples).**

| | Women | | Men | |
|---|---|---|---|---|
| | **Mean** | **Range** | **Mean** | **Range** |
| Age, years | 28 | 20-43 | 31 | 21-58 |
| Number of pregnancies | 3 | 1–7 | -- | -- |
| | **N** | **%** | **N** | **%** |
| Education level | | | | |
| None | 4 | 15% | 0 | 0% |
| Primary | 17 | 63% | 23 | 85% |
| High school | 6 | 22% | 4 | 15% |
| Received FP education/counseling* | 25 | 93% | 20 | 74% |
| Discussed short-term methods | 22 | 81% | 16 | 59% |
| Discussed implants | 22 | 81% | 13 | 48% |
| Discussed IUD | 5 | 19% | 2 | 7% |
| Received PPFP education/counseling | 23 | 85% | 17 | 63% |
| Know importance of PPFP | 26 | 96% | 19 | 70% |
| Women with concerns about side effects | 16 | 60% | -- | -- |
| Concerns about cost | 4 | 16% | -- | -- |
| Add cost and time | 6 | 22% | -- | -- |
| Religion barrier | 9 | 33% | -- | -- |
| Strategies to encourage women to consider or obtain PPFP | | | | |
| Educated the couple together | 3 | 11% | -- | -- |
| Involvement of male | 11 | 41% | -- | -- |

*17 couples (63%) received joint FP counseling/education; providers suggested use of implant to 20 couples (74%) and use of PPIUD to 11 (41%) couples, IUD: intrauterine device, PPFP: postpartum family planning.

## Access to PPFP counseling

Most women had received previous family planning educational counseling in a health center, and many men reported having had previous family planning educational counseling in a health center during ANC. Men and women who had not received family planning educational counseling in a health center reported hearing about family planning on the radio. One participant who did not receive PPFP educational counseling said, "**No, I had never heard about PP implants from a health professional, but I did read or get some information on them, such as that if a woman chooses PP implants and experiences negative effects, she may return to the health center and switch to another procedure.**"

Most couples had received family planning educational counseling together, primarily in ANC, and one couple reported receiving family planning education from a CHW during a village meeting. Most female participants had received PPFP education at health facilities after delivery, and one female participant said that she had heard about PPFP for the first time during her most recent delivery despite previous health facility deliveries.

Most women who had received family planning education had heard about implants and short-term methods.

## Knowledge about PPFP benefits

Almost all female participants thought PPFP was important, and one said that "**PPFP assists women in recovering and gaining the ability to work, contribute to family income, and alleviate poverty. It aids in birth spacing and gives women confidence in preventing unwanted pregnancy.**" More than half male participants had received PPFP

education or counseling and were aware of the importance of PPFP: "**The wife can conceive a few months after giving birth, and if she uses the PPFP methods, there is no concern of an unintended pregnancy.**"

## PPFP knowledge limitations

The most commonly suggested PPFP methods by providers were pills, injectables, and implants.

For many couples, postpartum implants were an option that was suggested by providers, and couples knew why it was important to uptake postpartum implants after delivery. For example, "**It prevents a woman from becoming pregnant while her child is still young and permits her to become pregnant when the youngest has grown up; it is easy to be removed; there are 2 options for 3 and 5 years; it helps in child spacing; and it helps for the wellbeing of the mother and the child.**" A few couples reported ever being counseled about the IUD and said PPIUD was an option that was discussed with them by a provider. One participant stated about the PPIUD: "**I just know it exists because some of my friends use it**", another participant said that "**I had never heard of PPIUD except from reading about it on hospital posters and hearing about it on the radio, and I had never discussed it with anyone who used it.**"

Some men asked questions about how IUDs and implants work so that they could explain them to their wives. One man asked, "**You said that there is a PPIUD; explain to us how it can be used. Is it like other methods, like pills or injectables?**", "**And I also need more explanation on PPIUD.**"

More generally, another participant said, "**We are mostly concerned with a lack of insufficient information on how family planning methods work.**" Other participants raised concerns about side effects and rumors heard from the community: "**I heard that the implant can move and go to other places of the body, but I'm not sure if that can harm the woman.**" and "**I would like to know if this is true; when someone uses an implant and has amenorrhea, can this cause tumors? I was using an implant and removed it due to fear of developing tumors.**" Some participants reported that the IUD bothers the male partner during sexual intercourse.

Participants suggested receiving comprehensive explanations about family planning methods before provision: "**The provider must explain the advantages and disadvantages of each family planning method before providing it. This may help the client choose the method when well aware of its side effects.**" Another suggested strategy was to start the family planning conversation early during pregnancy: "**Family planning discussions should start in the family before coming to the health facility in order to avoid making bad decisions at the time of provision after delivery. Discussing early on would allow them to check through the pros and cons of family planning methods and reach the facility ready to uptake.**"

## Concerns about PPFP

More than half participants expressed concern about the side effects of the implant, including bleeding, dizziness, weight loss or gain, backache and headache, and genital infections. One participant said that "**sometimes FP methods cause side effects that prevent people from taking them in the future.**"

A few participants expressed concerns about cost and time. One participant stated, "**It takes our time. It is not easy to convince a provider to remove it when you want to. Sometimes, it requires going to a private clinic for removal, which is expensive,**" and "**there is also a lack of health insurance in some families, which prevents them from taking FP.**" Another participant noted that "**it is free to access family planning services at the health center immediately after delivery; there is not much time spent receiving family planning methods. The problem is when you want to remove the family planning method.**"

Additionally, a few men were concerned about the potentially high cost spent on medical management of side effects.

Religion was occasionally expressed as a barrier: "**Most religious people do not support family planning; they said, this is to kill children inside the womb, and we are asking ourselves how we killed someone who does not yet exist. They promote traditional methods, which are not sins. People decided to take a risk and use the**

modern family planning method. **Using the family planning method is regarded as a sin (killing humans), which contradicts God's advice: "Multiply your descendants until you become as numerous as the sands on the sea"** Additionally, a few male participants said that some people think that family planning is for poor families who cannot afford to take care of their children.

## Strategies to engage male partners in PPFP

Some participants suggested PPFP education be provided to couples and that men could be approached for family planning education by sending a CHW to invite the couple to receive PPFP education together at home, during community gatherings, at church, during ANC visits, or at delivery. Involving male partners was viewed by some participant as very important. Participants said men should have a significant role in family planning decisions: "**Men need to understand family planning in order to support their wives; they have power or influence during the decision-making process of family planning; once they are against family planning, it is a challenge; and what is missing are promotional talks; and male partners are not exposed enough to health promotion. I am confident that men understand better once educated, and the decisions of family planning come from both partners. Discussing early on would allow them to check through the pros and cons of family planning methods and reach the facility ready to uptake.**" One participant noted that "**one partner refuses to proceed alone with getting the PPFP method, or when the woman takes the PPFP without husband approval, when side effects come, it creates misunderstanding in the couple.**"

## Perceptions on using PPFP testimonies during FP education sessions

All participants thought it would be helpful to receive testimonials from peers or women who are satisfied with their PPFP methods; one woman noted that "**listening to happy clients who talk about the positive impact of using PPFP would encourage people to uptake these methods**" and "**hearing testimonials reinforce hope and influence family planning decisions.**" One participant who received such a testimonial previously said that "**it would be helpful, and when I was a single, a woman who had planned successfully another pregnancy after nine years inspired me to use PPFP.**" However, there was one participant who noted: "**If I hear testimonials from unhappy clients, then I wouldn't use them at all, and I would discourage others from using them as well.**"

## Discussion

Overall, key findings from our study are the need to enhance PPFP training and provision in rural areas, especially for PPIUD, and common knowledge gaps and concerns about side effects emphasized in FGDs.

PPFP could have a major live-saving impact for rural Rwandans, where 80% of Rwandans live in rural areas [17]. Though desires for large family sizes in rural areas have decreased due to government educational campaigns on health, poverty, and land scarcity [18], rural total fertility remains higher despite the mean desired number of children being similar to urban areas. Though rural areas have seen a 39% increase in FP use over the past decade [17,19], rural women are more likely to report unmet FP need for birth limiting and lower IUD use than urban women (0.7% vs 3%) [17]. Additionally, inter-pregnancy intervals are shorter and rates of infant, under-5 [17], and maternal [20] mortality are higher in rural vs urban women.

Our study found that PPFP services are being offered in rural health centers and hospitals, though few nurses and midwives are trained in PPIUD provision. Physicians at hospitals were more frequently trained in PPIUD insertion (32%) than implant insertion (14%). Over half of CHW had received training in PPFP promotion. However, not all essential materials to facilitate the provision of PPLARC methods were available at the health centers and hospitals. This contrasts with a study conducted in Bangladesh, which showed that rural areas had all the necessary materials to provide LARC compared to urban areas [44]. Few nurses were trained in LARC provision, and there was a lack of PPFP provision materials, similar to our previous work in 2016 conducted in 30 rural health settings in Rwanda [29].

In terms of PPLARC method provision, we found that all health facilities provided PP implants, but no PPIUD methods were offered at health centers despite available stock for any FP methods. This aligns with another study showing that implants were the most commonly provided LARC method in Rwanda and FP initiators tended to prefer implants [29]. Low PPIUD provision may be attributed to limited provider training, ineffective counseling, lack of client knowledge about PPIUD, and misinformation. Notably, PPIUD is the most effective and cost-effective method for preventing pregnancy with few side effects compared to implants [8–11].

The findings of this study also indicated that most pregnant couples had received family planning and PPFP education and counseling, and they understood the importance of FP and PPFP. A study conducted in rural Tanzania revealed that low levels of family planning literacy among women can significantly hinder family planning uptake [45]. Correspondingly, research findings from two public hospitals in Rwanda indicated that marital status and active spousal involvement play a crucial role in positively influencing immediate family planning utilization [46]. This is largely due to the promotion of family planning and PPFP promotion during antenatal services at health centers, and in Rwanda, during the first ANC visit, both partners attended the ANC visit together and received PPFP education. Between 2010 and 2020, there was an increase in modern family planning use in Rwanda, which is attributed to changes in knowledge and attitudes toward family planning among rural women [47]. Sustained family planning use among rural Rwandan women has been associated with an increase in patient-centered counseling on potential side effects and peer counseling [47].

This study identified a lack of knowledge about how the family planning methods work and their side effects. However, there is an opportunity to enhance family planning and PPFP education or counseling in rural health facilities. Notably, a majority of rural women attend at least one ANC visit (98%), deliver in a health facility (91%), and obtain all eight basic infant vaccinations (95%), making these touchpoints highly viable service delivery points for PPFP promotion and provision [17].

Most of the couples reported that the postpartum implant was the method most frequently discussed with them by providers, and this may be due to the limited number of providers who are trained to provide the PPIUD method. This finding aligns with the results of a study conducted by Alexandra et al., which showed that a lack of provider training is a major barrier to LARC provision [48]. Study participants also indicated that male partners play a significant role in PPFP method uptake decisions. Our PPFP work in Kigali demonstrated that male partners can be effectively engaged by providing educational counseling materials designed for couples at first ANC visits, 80% of which are attended by male partners in Rwanda [23,39,49–51]. Additionally, health posts are the primary healthcare facilities closest to the community were identified as key points for strengthening the capacity and training of healthcare providers for increased family planning and PPFP promotion and provision.

The ongoing PPFP intervention taking place in Kigali, Icyemezo Cyacu ('Our Decision' in Kinyarwanda), which was informed by women and their male partners, providers, and stakeholders, aims to guide in the sustainability and cost-effectiveness of PPFP in urban areas [40,52]. The focus group participants requested FP education for couples to increase uptake mirroring findings from a previous study showing that integration of FP counseling in couples HIV counselling and testing increases the adoption of LARC methods [40,52].

According to the *Scaling Up Immediate Postpartum Family Planning Services in Rwanda* report [53], the rollout and expansion of PPFP services began in 2016 with capacity building in selected districts, followed by nationwide scaling in all districts of Rwanda by 2020, resulted in increased availability and accessibility of PPFP services at all levels. PPFP can also help Rwanda achieve its goal of reducing unmet need for family planning from 13.8% in 2024/25–8% by 2029 [54].

## Limitations

The results of this study should be interpreted recognizing the following limitations. The study cannot be generalized beyond the couple as it did not include data from women or men alone. Our deductive thematic analysis which focused on pre-specified themes emergent in prior studies means it is possible that themes which would have emerged in an

inductive approach were not identified. Although, the study team is experienced in designing FG guide and conducting interviews, social approval bias when collecting self-reported information is possible. Finally, we did not collect data on biases or assumptions held by the CFHR interviewers which could have influenced the discussion.

## Conclusions and recommendations

Our study highlights the need to enhance PPFP training and provision in rural areas, especially for PPIUD, and confirmed a low postpartum LARC uptake. Knowledge gaps and concerns about side effects were emphasized in FGDs. The Rwandan government and its partners have made efforts to improve accessibility and availability of all family planning methods, resulting in increased family planning uptake nationwide. To achieve Rwanda's FP2030 commitment to expand access to FP for women and girls throughout the country and to reduce the unmet need for FP/PPFP, we recommend FP stakeholders to encourage facilities to deliver comprehensive FP education information including IUD/PPIUD, ensure regular FP provider training to factor in IUD/PPIUD insertion skills and staff turn-over, and allow mentorships program to address regularly issues emerging from facility monthly reports/needs. PPFP educational materials should be revised to ensure a comprehensive, inclusive, interactive, and friendly delivery of FP information by providers to clients. Additionally, PPFP demand creation strategies should be tailored for rural communities and, rural health posts could be explored as potential locations for PPFP counseling. Future studies should also investigate other factors that would sustain uptake and a delivery of full range of PPFP methods overtime despite regular provider training and provision of FP material/ equipment.

## Supporting information

**S1 Fig. Code tree guiding focus group themes and sub-themes.**
(DOCX)

**S1 Appendix. COREQ (COnsolidated criteria for REporting Qualitative research) Checklist.**
(DOCX)

## Author contributions

**Conceptualization:** Claudine Umuhoza, Rosine Ingabire, Kristin M. Wall.

**Data curation:** Claudine Umuhoza, Mame Diarra Niang, Rachel Parker.

**Formal analysis:** Claudine Umuhoza, Rachel Parker.

**Funding acquisition:** Mame Diarra Niang, Kristin M. Wall.

**Investigation:** Claudine Umuhoza, Mame Diarra Niang, Rosine Ingabire, Azhar Nizam, Jessica Sales, Susan Allen, Etienne Karita.

**Methodology:** Claudine Umuhoza, Rosine Ingabire, Amelia Mazzei, Rachel Parker, Lisa Haddad, Phaedra Corso, Susan Allen, Etienne Karita.

**Project administration:** Claudine Umuhoza, Mame Diarra Niang, Rosine Ingabire, Amelia Mazzei, Amanda Tichacek, Susan Allen, Etienne Karita.

**Resources:** Susan Allen, Kristin M. Wall, Etienne Karita.

**Supervision:** Claudine Umuhoza, Rosine Ingabire, Julien Nyombayire, Amelia Mazzei, Amanda Tichacek, Susan Allen, Etienne Karita.

**Writing – original draft:** Claudine Umuhoza.

**Writing – review & editing:** Claudine Umuhoza, Mame Diarra Niang, Rosine Ingabire, Julien Nyombayire, Amelia Mazzei, Rachel Parker, Amanda Tichacek, Azhar Nizam, Jessica Sales, Lisa Haddad, Phaedra Corso, Susan Allen, Kristin M. Wall, Etienne Karita.

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
