## [Decision Letter · Decision Letter 0]

20 Nov 2024

PONE-D-24-28650
Implementing postpartum family planning services in rural Rwanda: a mixed-methods study
PLOS ONE

Dear Dr. Wall,

Thank you for submitting your manuscript to PLOS ONE. After careful consideration, we feel that it has merit but does not fully meet PLOS ONE’s publication criteria as it currently stands. Therefore, we invite you to submit a revised version of the manuscript that addresses the points raised during the review process.

**According to the outcome of the peer review, almost all sections requires major revisions.**

We look forward to receiving your revised manuscript.

Kind regards,

**Dr Syed Khurram Azmat**, PhD, MPH, MD

Academic Editor

PLOS ONE

**Journal requirements:**

**Additional Editor Comments:**

Major revision is mandatory.

Reviewers' comments:

Reviewer's Responses to Questions

**Comments to the Author**

1. Is the manuscript technically sound, and do the data support the conclusions?

Reviewer #1: No

Reviewer #2: Partly

2. Has the statistical analysis been performed appropriately and rigorously? 

Reviewer #1: No

Reviewer #2: No

3. Have the authors made all data underlying the findings in their manuscript fully available?

Reviewer #1: Yes

Reviewer #2: No

4. Is the manuscript presented in an intelligible fashion and written in standard English?

Reviewer #1: No

Reviewer #2: No

5. Review Comments to the Author

**Reviewer #1:** Thank you for the opportunity to review this manuscript. The study addresses an important topic concerning Postpartum Family Planning (PPFP) and utilizes both quantitative and qualitative data sources. However, the manuscript requires major revisions.

General comments

The methods and results sections need thorough revision, particularly regarding the qualitative study, which is poorly described. I recommend adhering to the COREQ guidelines to enhance the quality of the qualitative reporting. Additionally, please consult textbooks on mixed methods research, such as Creswell’s work, to provide readers with a better understanding of the different types of mixed methods studies and how to report them.

Specific comments

#### Abstract

- Methods: Clearly specify the analytical techniques used for the quantitative part of the study.

- Results: Start by stating the total number of data points analyzed. Distinguish clearly between quantitative and qualitative results. While the abstract currently summarizes quantitative findings, please specify the key themes and main takeaways from the qualitative findings without including specific values.

#### Background

Although this section provides necessary context, it lacks a rationale for the study's importance. Please revise to highlight research gaps and the implications of your findings, creating a narrative that connects to the study objectives.

The last paragraph of the introduction should clearly articulate the research objectives. It currently summarizes the methods rather than stating the study’s aims.

#### Methods

- **Study Design**: Specify the type of mixed methods used (sequential explanatory/exploratory or concurrent design). If sequential, indicate whether quantitative or qualitative data collection precedes the other. (Refer to Designing and Conducting Mixed Methods Research

Book by John W. Creswell and Vicki L. Plano Clark)

- **Study Sites and Recruitment**: Clarify how and who determined that the facilities were high-volume and accessible. What criteria were used? Was it based on annual patient flow, formal reports, or insights from key stakeholders?

- **Sampling**: Specify the sampling methods for both quantitative and qualitative data collection. Was convenience sampling used for quantitative data or was it sourced from a database? Was purposive sampling employed for qualitative data?

- **Qualitative Data Collection and Analysis**: For Focus Group Discussions (FGDs), please clarify the data analysis process. Specify if coding was conducted and whether a thematic analysis was used.

#### Results

Clearly separate the findings into quantitative and qualitative sections. Explain how the results from the two study designs intersect—whether this occurs during data analysis or interpretation.

The qualitative findings are inadequately reported. Please adhere to

Consolidated criteria for reporting qualitative research (COREQ)

guidelines and ensure the following:

- State whether saturation of information was reached.

- Provide a code tree with clear themes, sub-themes, and representative quotes for each theme.

- Ensure the voices of both majority and minority participants, including both negative and positive perspectives, are represented.

Please also submit the COREQ checklist as a supplemental file in your next revision

#### Discussion

Begin this section by summarizing the key findings and the main messages readers should retain.

The current discussion is overly simplistic. Please elaborate on each key finding, including its implications, and compare these results with studies from other countries to provide a global perspective.

Additionally, a limitations section is missing and should be included.

Please also add a recommendations section, rooted in key findings, that addresses potential policy implications.

#### Conclusion

Revise the conclusion to remind readers of the key findings and recommendations in a concise manner.

Thank you for considering these suggestions to enhance the manuscript's clarity and impact.

**Reviewer #2: **The research focuses on a very important topic, but unfortunately the methodology (sampling, inclusion, & exclusion criteria and narration of mix methodology) as well as the results require narration with a lot more clarity. Moreover, consistency and flow of the narration needs improvement. Therefore would advise a major review.

6. PLOS authors have the option to publish the peer review history of their article (what does this mean?). If published, this will include your full peer review and any attached files.

Reviewer #1: No

Reviewer #2: No

---

## [Author Response · Author response to Decision Letter 1]

20 May 2025

Reviewers' comments:

5. Review Comments to the Author

Reviewer #1: Thank you for the opportunity to review this manuscript. The study addresses an important topic concerning Postpartum Family Planning (PPFP) and utilizes both quantitative and qualitative data sources. However, the manuscript requires major revisions.

General comments

The methods and results sections need thorough revision, particularly regarding the qualitative study, which is poorly described. I recommend adhering to the COREQ guidelines to enhance the quality of the qualitative reporting. Additionally, please consult textbooks on mixed methods research, such as Creswell’s work, to provide readers with a better understanding of the different types of mixed methods studies and how to report them.

Response: We appreciate these suggestions and have adhered to COREQ; the COREQ checklist is now included as an appendix. We have also expanded our qualitative data analysis methods to provide more details, and have revised our results to align with COREQ and guidance on concurrent mixed methods designs from Creswell & Plano Clark, 2011.

Specific comments

#### Abstract

- Methods: Clearly specify the analytical techniques used for the quantitative part of the study.

Response: We now specify that the quantitative data were analyzed descriptively.

- Results: Start by stating the total number of data points analyzed. Distinguish clearly between quantitative and qualitative results. While the abstract currently summarizes quantitative findings, please specify the key themes and main takeaways from the qualitative findings without including specific values.

Response: We now distinguish the qualitative and the quantitative data in the abstract, detail the total number of data points analyzed for each analysis, and summarize the key qualitative themes.

#### Background

Although this section provides necessary context, it lacks a rationale for the study's importance. Please revise to highlight research gaps and the implications of your findings, creating a narrative that connects to the study objectives.

The last paragraph of the introduction should clearly articulate the research objectives. It currently summarizes the methods rather than stating the study’s aims.

Response: The last paragraph of the introduction was revised to highlight the gaps in current knowledge that this study addresses.

#### Methods

- **Study Design**: Specify the type of mixed methods used (sequential explanatory/exploratory or concurrent design). If sequential, indicate whether quantitative or qualitative data collection precedes the other. (Refer to Designing and Conducting Mixed Methods Research

Book by John W. Creswell and Vicki L. Plano Clark)

Response: We have expanded our qualitative data analysis methods to provide more details, and have revised our results to align with COREQ and guidance on concurrent mixed methods designs from Creswell & Plano Clark, 2011.

- **Study Sites and Recruitment**: Clarify how and who determined that the facilities were high-volume and accessible. What criteria were used? Was it based on annual patient flow, formal reports, or insights from key stakeholders?

Response: We now clarify how facilities were selected for inclusion in the methods section.

- **Sampling**: Specify the sampling methods for both quantitative and qualitative data collection. Was convenience sampling used for quantitative data or was it sourced from a database? Was purposive sampling employed for qualitative data?

Response: We now clarify how facilities were selected for inclusion in the methods section. We also clarify that convenience sampling was employed for recruitment of FGD participants who were invited from antenatal care services.

- **Qualitative Data Collection and Analysis**: For Focus Group Discussions (FGDs), please clarify the data analysis process. Specify if coding was conducted and whether a thematic analysis was used.

Response: The qualitative data analysis methods have been more thoroughly described. As described in the text, deductive thematic analysis was used, without coding.

#### Results

Clearly separate the findings into quantitative and qualitative sections. Explain how the results from the two study designs intersect—whether this occurs during data analysis or interpretation.

Response: The results from qualitative and quantitative are now presented in separate sections. The quantitative and qualitative data converge at the interpretation stage.

The qualitative findings are inadequately reported. Please adhere to

Consolidated criteria for reporting qualitative research (COREQ)

guidelines and ensure the following:

- State whether saturation of information was reached.

- Provide a code tree with clear themes, sub-themes, and representative quotes for each theme.

- Ensure the voices of both majority and minority participants, including both negative and positive perspectives, are represented.

Please also submit the COREQ checklist as a supplemental file in your next revision

Response: Each of these items have now been described and/or reported in the manuscript. The COREQ checklist is completed and attached.

#### Discussion

Begin this section by summarizing the key findings and the main messages readers should retain.

Response: The discussion now begins with a summary of the key findings/main messages.

The current discussion is overly simplistic. Please elaborate on each key finding, including its implications, and compare these results with studies from other countries to provide a global perspective.

Response: We now further contextualize our findings in light of prior literature in the in the region.

Additionally, a limitations section is missing and should be included.

Response: A limitations section has been added.

Please also add a recommendations section, rooted in key findings, that addresses potential policy implications.

#### Conclusion

Revise the conclusion to remind readers of the key findings and recommendations in a concise manner.

Response: The conclusions section has been revised to reiterate the key findings and to make recommendations rooted in findings.

Thank you for considering these suggestions to enhance the manuscript's clarity and impact.

Reviewer #2: The research focuses on a very important topic, but unfortunately the methodology (sampling, inclusion, & exclusion criteria and narration of mix methodology) as well as the results require narration with a lot more clarity. Moreover, consistency and flow of the narration needs improvement. Therefore would advise a major review.

Response: We appreciate this comment and hope that the revisions and edits noted above and within the manuscript address these issues.

6. PLOS authors have the option to publish the peer review history of their article (what does this mean?). If published, this will include your full peer review and any attached files.

Do you want your identity to be public for this peer review? For information about this choice, including consent withdrawal, please see our Privacy Policy.

Reviewer #1: No

Reviewer #2: No

---

## [Decision Letter · Decision Letter 1]

26 Aug 2025

PONE-D-24-28650R1
Implementing postpartum family planning services in rural Rwanda: a mixed-methods study
PLOS ONE

Dear Dr. Wall,

Thank you for submitting your manuscript to PLOS ONE. After careful consideration, we feel that it has merit but does not fully meet PLOS ONE’s publication criteria as it currently stands. Therefore, we invite you to submit a revised version of the manuscript that addresses the points raised during the review process.

Based on the outcome of peer-review, the manuscript requires to undergo major revisions before the journal can further considers it.

We look forward to receiving your revised manuscript.

Kind regards,

Syed Khurram Azmat, PhD, MPH, MD

Academic Editor

PLOS ONE

Journal Requirements:

Reviewers' comments:

Reviewer's Responses to Questions

**Comments to the Author**

1. If the authors have adequately addressed your comments raised in a previous round of review and you feel that this manuscript is now acceptable for publication, you may indicate that here to bypass the “Comments to the Author” section, enter your conflict of interest statement in the “Confidential to Editor” section, and submit your "Accept" recommendation.

Reviewer #1: (No Response)

Reviewer #3: All comments have been addressed

2. Is the manuscript technically sound, and do the data support the conclusions?

Reviewer #1: Partly

Reviewer #3: Partly

3. Has the statistical analysis been performed appropriately and rigorously? 

Reviewer #1: Yes

Reviewer #3: No

4. Have the authors made all data underlying the findings in their manuscript fully available?

Reviewer #1: Yes

Reviewer #3: Yes

5. Is the manuscript presented in an intelligible fashion and written in standard English?

Reviewer #1: Yes

Reviewer #3: Yes

6. Review Comments to the Author

Reviewer #1: Thank you for the opportunity to review the revised manuscript. The manuscript has improved remarkably, and the authors have addressed most of the previous comments. However, the research implications remain unaddressed, and the recommendation section has been entirely omitted. This is a significant oversight. Additionally, the manuscript reflects a lack of experience and understanding in reporting qualitative research, particularly in the presentation of qualitative findings, which require major revision.

Below are other specific comments for your consideration:

Abstract

Methods – Avoid presenting numerical data in the abstract for qualitative research. Instead, summarize the key themes that emerged from the study and highlight the main takeaway message. Simply stating, “The majority of the 27 couples recruited for FGDs (93% of women, 74% of men) had received family planning educational counseling though less than half (41%) of couples reported learning about the PPIUD” does not adequately reflect the richness and depth of the qualitative findings. This approach oversimplifies the data and misses the nuanced perspectives shared by participants. Emphasis should be placed on the insights gained about participants’ experiences, perceptions, and barriers related to family planning and PPIUD counseling.

Introduction

Since the introduction mentions both the cost of contraceptives and Rwanda’s promotion of postpartum family planning (PPFP) and long-acting reversible contraception (LARC), please include a brief context on how these services are accessed in practice. Are they freely available through the public health system in Rwanda? Adding one or two sentences—ideally with a valid reference—about the availability and affordability of PPFP and LARC would provide a more complete picture. This addition could be placed in either the second or third paragraph of the introduction without disrupting the narrative flow.

Implication of the study

While the authors responded that they have added the implications of the study, I do not see these changes reflected in the current version of the introduction. The section still lacks a clear articulation of the research gap. Although structural barriers are mentioned, it remains unclear whether similar studies on postpartum family planning (PPFP) or PPIUD have been conducted in Rwanda before. If such studies exist, what gap does this research address? What new insights does it contribute to the existing evidence base? If no similar studies exist, why is this study needed, and how can it inform policy, practice, or future research? I recommend elaborating on these points to clearly establish the significance and contribution of this study. Please add this to the last paragraph of the introduction section, just before the aims of the study.

Methods

Line 141 – The sampling method used for the FGDs appears to be purposive rather than convenience sampling. There are important nuances between these two approaches: purposive sampling involves deliberately selecting participants based on specific characteristics relevant to the research question, while convenience sampling relies on accessibility and availability. Please review the definitions of both methods and revise the manuscript accordingly if purposive sampling was indeed used.

Line 173 – correct the word “propose” to “purpose”.

Line 197 – In qualitative research, while we may have a general idea of the approximate sample size needed, it is not rigidly predetermined as in quantitative studies. Since your data collection continued until the point of saturation, it would be more appropriate to simply state that. Please revise the sentence “The sample size was pre-specified to be sufficient to reach saturation based on prior guidelines for saturation”—as this is technically not accurate for qualitative research. A more appropriate phrasing could be: “Data collection continued until thematic saturation was reached, in line with qualitative research principles.”

Qualiative findingss

The reporting of qualitative findings still requires improvement.

Please avoid presenting percentages when describing responses. In qualitative research, we focus on the depth and meaning of participants’ experiences rather than quantifying how many said what. Using terms like “majority” or providing percentages undermines the interpretive nature of qualitative data.

Moreover, the code tree or thematic structure is missing, which is an important component recommended by the COREQ (Consolidated Criteria for Reporting Qualitative Research) guidelines. Please include a table outlining the main themes and their corresponding subthemes or categories. Even if the themes were predefined, the findings should still be clearly structured under each one to improve readability and coherence. At present, the lack of an organized thematic framework makes the results difficult to follow.

Since you’ve indicated that the Theory of Planned Behavior guided the study, consider aligning the themes with its core constructs. This would not only enhance theoretical consistency but also strengthen the interpretive value of your findings.

The predefined themes outlined in the manuscript provide limited insight into the how and why behind participants’ experiences and thus offer very little depth. To fully capture the richness of the qualitative data, the analysis should move beyond superficial descriptions and explore underlying meanings, processes, and contextual factors that shape participants’ perspectives.

The reporting of qualitative findings requires major revision to adequately reflect the depth, complexity, and richness of the data.

Discussion-limitations

Lines 437–439 – This statement does not represent a true limitation. The note that illustrative quotes cannot be linked to participants’ age, number of pregnancies, children, or previous use of PPFP is unnecessary. In qualitative research, it is common practice to provide contextual information such as participant gender, age group, or parity when presenting quotes, without necessarily linking every response to detailed demographic data. I suggest revising this section to reflect that quotes can still be meaningfully contextualized by key participant characteristics like gender and parity, which would enrich the findings rather than constitute a limitation.

Please revise the limitations section. The points currently listed do not represent valid limitations of the study. Instead, consider discussing methodological constraints such as reliance on existing quantitative databases with limited variables, potential recall bias, or challenges related to the study timeline and data collection period. Reflecting on these practical and contextual limitations will provide a more meaningful and balanced assessment of the study’s scope and potential weaknesses.

Recommendations

The manuscript did not address my earlier comment regarding the recommendations section, which is currently missing. Please include a dedicated recommendations section that clearly highlights how the findings can inform policymakers, practitioners working in postpartum family planning (PPFP), and future research in this area. Ensure that these recommendations are directly rooted in and supported by the study’s findings to enhance their relevance and practical utility.

Reviewer #3: This paper is interesting, but has a fair bit of revision necessary in order to be publishable.

As a minor point, there are too many writing and grammatical errors, particularly numerical inconsistencies (six instead of 6).

More substantively, it would be helpful to know more about these two regions of Rwanda. How to they compare to others in terms of population density, prior FP use, ethnic diversity (e.g. refugee populations from surrounding nations). Outside of Kigali much of the nation is rural, but there is still demographic diversity so more information on why these two regions are selected (beyond convenience) would be helpful to contextualize the results.

The quantitative analysis is somewhat limited in terms of comparisons. If some clinics/hospitals/health posts have limited capabilities for PPFP, is that consistent with other regions? Of course 100% is ideal, but what is realistic, and how far are these numbers from the realistic goal?

Qualitative focus groups should include more information as to how data was collected. Were husbands and wives in the same room at the same time? Could this influence responses? It is a real strength of this paper to include male voices in family planning conversations, but much more depth and details is needed to explain the process through which this data was collected, and the potential sources of bias and error in a highly patriarchal culture.

Most importantly, what information are we gleaning from this study beyond the idea that more resources, training, and information would improve PPFP outcomes. What can we learn from this that would help Rwandan communities specifically and public health scholars more generally?

7. PLOS authors have the option to publish the peer review history of their article (what does this mean?). If published, this will include your full peer review and any attached files.

Reviewer #1: No

Reviewer #3: No

---

## [Author Response · Author response to Decision Letter 2]

27 Oct 2025

PONE-D-24-28650R1

Implementing postpartum family planning services in rural Rwanda: a mixed-methods study

PLOS ONE

Dear Dr. Wall,

Thank you for submitting your manuscript to PLOS ONE. After careful consideration, we feel that it has merit but does not fully meet PLOS ONE’s publication criteria as it currently stands. Therefore, we invite you to submit a revised version of the manuscript that addresses the points raised during the review process.

Based on the outcome of peer-review, the manuscript requires to undergo major revisions before the journal can further considers it.

We look forward to receiving your revised manuscript.

Kind regards,

Syed Khurram Azmat, PhD, MPH, MD

Academic Editor

PLOS ONE

Journal Requirements:

Reviewers' comments:

Reviewer's Responses to Questions

Comments to the Author

1. If the authors have adequately addressed your comments raised in a previous round of review and you feel that this manuscript is now acceptable for publication, you may indicate that here to bypass the “Comments to the Author” section, enter your conflict of interest statement in the “Confidential to Editor” section, and submit your "Accept" recommendation.

Reviewer #1: (No Response)

Reviewer #3: All comments have been addressed

2. Is the manuscript technically sound, and do the data support the conclusions?

Reviewer #1: Partly

Reviewer #3: Partly

3. Has the statistical analysis been performed appropriately and rigorously?

Reviewer #1: Yes

Reviewer #3: No

4. Have the authors made all data underlying the findings in their manuscript fully available?

Reviewer #1: Yes

Reviewer #3: Yes

5. Is the manuscript presented in an intelligible fashion and written in standard English?

Reviewer #1: Yes

Reviewer #3: Yes

6. Review Comments to the Author

Reviewer #1: Thank you for the opportunity to review the revised manuscript. The manuscript has improved remarkably, and the authors have addressed most of the previous comments. However, the research implications remain unaddressed, and the recommendation section has been entirely omitted. This is a significant oversight. Additionally, the manuscript reflects a lack of experience and understanding in reporting qualitative research, particularly in the presentation of qualitative findings, which require major revision.

Below are other specific comments for your consideration:

Abstract

Methods – Avoid presenting numerical data in the abstract for qualitative research. Instead, summarize the key themes that emerged from the study and highlight the main takeaway message. Simply stating, “The majority of the 27 couples recruited for FGDs (93% of women, 74% of men) had received family planning educational counseling though less than half (41%) of couples reported learning about the PPIUD” does not adequately reflect the richness and depth of the qualitative findings. This approach oversimplifies the data and misses the nuanced perspectives shared by participants. Emphasis should be placed on the insights gained about participants’ experiences, perceptions, and barriers related to family planning and PPIUD counseling.

Response: We have reviewed the results section of the abstract, and the qualitative findings are now presented as themes highlighting the main takeaway messages of our study.

Introduction

Since the introduction mentions both the cost of contraceptives and Rwanda’s promotion of postpartum family planning (PPFP) and long-acting reversible contraception (LARC), please include a brief context on how these services are accessed in practice. Are they freely available through the public health system in Rwanda? Adding one or two sentences—ideally with a valid reference—about the availability and affordability of PPFP and LARC would provide a more complete picture. This addition could be placed in either the second or third paragraph of the introduction without disrupting the narrative flow.

Response: We have briefly described how affordable and available family planning (FP) services are in Rwanda in the 3rd paragraph. Briefly, FP services are provided in both public and private settings and are covered by public health insurance which provides essentially universal healthcare. Refills of short term methods are provided free of charge in the community by community health workers.

Implication of the study

While the authors responded that they have added the implications of the study, I do not see these changes reflected in the current version of the introduction. The section still lacks a clear articulation of the research gap. Although structural barriers are mentioned, it remains unclear whether similar studies on postpartum family planning (PPFP) or PPIUD have been conducted in Rwanda before. If such studies exist, what gap does this research address? What new insights does it contribute to the existing evidence base? If no similar studies exist, why is this study needed, and how can it inform policy, practice, or future research? I recommend elaborating on these points to clearly establish the significance and contribution of this study. Please add this to the last paragraph of the introduction section, just before the aims of the study.

Response: We have added a paragraph in the introduction highlighting the impact of our study. While the few FP studies conducted in rural Rwanda have confirmed the low uptake of PPFP, there has not been an assessment of rural facility/staff readiness in offering PPFP services along with both women and their male partners perceptions of PPFP needs and services.

Methods

Line 141 – The sampling method used for the FGDs appears to be purposive rather than convenience sampling. There are important nuances between these two approaches: purposive sampling involves deliberately selecting participants based on specific characteristics relevant to the research question, while convenience sampling relies on accessibility and availability. Please review the definitions of both methods and revise the manuscript accordingly if purposive sampling was indeed used.

Response: We confirm that our sampling method used for the FGDs was convenience sampling. As explained from Line 145-157, the study team selected high volume, accessible and non-Catholic facilities in the two rural districts, and all adult pregnant women coming with their partner were recruited for FGDs from ANC services. No additional criteria were used to purposively select participants for focus groups.

Line 173 – correct the word “propose” to “purpose”.

Response: This type has been corrected.

Line 197 – In qualitative research, while we may have a general idea of the approximate sample size needed, it is not rigidly predetermined as in quantitative studies. Since your data collection continued until the point of saturation, it would be more appropriate to simply state that. Please revise the sentence “The sample size was pre-specified to be sufficient to reach saturation based on prior guidelines for saturation”—as this is technically not accurate for qualitative research. A more appropriate phrasing could be: “Data collection continued until thematic saturation was reached, in line with qualitative research principles.”

Response: Line 209 was updated accordingly and we removed the previous statement as suggested.

Qualiative findingss

The reporting of qualitative findings still requires improvement.

Please avoid presenting percentages when describing responses. In qualitative research, we focus on the depth and meaning of participants’ experiences rather than quantifying how many said what. Using terms like “majority” or providing percentages undermines the interpretive nature of qualitative data.

Moreover, the code tree or thematic structure is missing, which is an important component recommended by the COREQ (Consolidated Criteria for Reporting Qualitative Research) guidelines. Please include a table outlining the main themes and their corresponding subthemes or categories. Even if the themes were predefined, the findings should still be clearly structured under each one to improve readability and coherence. At present, the lack of an organized thematic framework makes the results difficult to follow.

Since you’ve indicated that the Theory of Planned Behavior guided the study, consider aligning the themes with its core constructs. This would not only enhance theoretical consistency but also strengthen the interpretive value of your findings.

The predefined themes outlined in the manuscript provide limited insight into the how and why behind participants’ experiences and thus offer very little depth. To fully capture the richness of the qualitative data, the analysis should move beyond superficial descriptions and explore underlying meanings, processes, and contextual factors that shape participants’ perspectives.

The reporting of qualitative findings requires major revision to adequately reflect the depth, complexity, and richness of the data.

Response: The code tree is now added, and the qualitative results have been rearranged based on themes emerging from the data.

Discussion-limitations

Lines 437–439 – This statement does not represent a true limitation. The note that illustrative quotes cannot be linked to participants’ age, number of pregnancies, children, or previous use of PPFP is unnecessary. In qualitative research, it is common practice to provide contextual information such as participant gender, age group, or parity when presenting quotes, without necessarily linking every response to detailed demographic data. I suggest revising this section to reflect that quotes can still be meaningfully contextualized by key participant characteristics like gender and parity, which would enrich the findings rather than constitute a limitation.

Please revise the limitations section. The points currently listed do not represent valid limitations of the study. Instead, consider discussing methodological constraints such as reliance on existing quantitative databases with limited variables, potential recall bias, or challenges related to the study timeline and data collection period. Reflecting on these practical and contextual limitations will provide a more meaningful and balanced assessment of the study’s scope and potential weaknesses.

Response: We do believe that by recruiting participants right after attending ANC visits where FP counseling messages are discussed that the study team has mitigated recall bias. We do however recognize that social desirability bias would be possible in this study. The limitations section has been revised accordingly.

Recommendations

The manuscript did not address my earlier comment regarding the recommendations section, which is currently missing. Please include a dedicated recommendations section that clearly highlights how the findings can inform policymakers, practitioners working in postpartum family planning (PPFP), and future research in this area. Ensure that these recommendations are directly rooted in and supported by the study’s findings to enhance their relevance and practical utility.

Response: The recommendations section was revised to include recommendations for stakeholders and practitioners as well as future investigations needed to sustain PPFP uptake overtime.

Reviewer #3: This paper is interesting, but has a fair bit of revision necessary in order to be publishable.

As a minor point, there are too many writing and grammatical errors, particularly numerical inconsistencies (six instead of 6).

Response: The manuscript was proofread, and grammatical errors were addressed.

More substantively, it would be helpful to know more about these two regions of Rwanda. How to they compare to others in terms of population density, prior FP use, ethnic diversity (e.g. refugee populations from surrounding nations). Outside of Kigali much of the nation is rural, but there is still demographic diversity so more information on why these two regions are selected (beyond convenience) would be helpful to contextualize the results.

Response: A short paragraph on the selection of the two rural district was added.

The quantitative analysis is somewhat limited in terms of comparisons. If some clinics/hospitals/health posts have limited capabilities for PPFP, is that consistent with other regions? Of course 100% is ideal, but what is realistic, and how far are these numbers from the realistic goal?

Qualitative focus groups should include more information as to how data was collected. Were husbands and wives in the same room at the same time? Could this influence responses? It is a re

---

## [Editor Report · Decision Letter 2]

25 Nov 2025

Implementing postpartum family planning services in rural Rwanda: a mixed-methods study

PONE-D-24-28650R2

Dear Dr. Wall,

We’re pleased to inform you that your manuscript has been judged scientifically suitable for publication and will be formally accepted for publication once it meets all outstanding technical requirements.

Kind regards,

Syed Khurram Azmat, PhD, MPH, MD

Academic Editor

PLOS ONE
---

## [Editor Report · Acceptance letter]

PONE-D-24-28650R2

PLOS One

Dear Dr. Wall,

I'm pleased to inform you that your manuscript has been deemed suitable for publication in PLOS One. Congratulations! Your manuscript is now being handed over to our production team.

Kind regards,

on behalf of

Dr. Syed Khurram Azmat

Academic Editor

PLOS One